# What Is *Candida* Doing in My Food? A Review and Safety Alert on Its Use as Starter Cultures in Fermented Foods

**DOI:** 10.3390/microorganisms10091855

**Published:** 2022-09-16

**Authors:** Gilberto Vinícius de Melo Pereira, Bruna Leal Maske, Dão Pedro de Carvalho Neto, Susan Grace Karp, Juliano De Dea Lindner, José Guilherme Prado Martin, Bianca de Oliveira Hosken, Carlos Ricardo Soccol

**Affiliations:** 1Department of Bioprocess Engineering and Biotechnology, Federal University of Paraná (UFPR), Curitiba 80230-901, PR, Brazil; 2Federal Institute of Education, Science and Technology of Paraná (IFPR), Londrina 86041-120, PR, Brazil; 3Department of Food Science and Technology, Federal University of Santa Catarina (UFSC), Florianópolis 88040-900, SC, Brazil; 4Microbiology Department, Federal University of Viçosa (UFV), Viçosa 36570-900, MG, Brazil

**Keywords:** fermented foods, *Candida* spp., food safety, wine, cocoa

## Abstract

The use of yeasts as starter cultures was boosted with the emergence of large-scale fermentations in the 20th century. Since then, *Saccharomyces cerevisiae* has been the most common and widely used microorganism in the food industry. However, *Candida* species have also been used as an adjuvant in cheese production or as starters for coffee, cocoa, vegetable, meat, beer, and wine fermentations. A thorough screening of candidate *Candida* is sometimes performed to obtain the best performing strains to enhance specific features. Some commonly selected species include *C. pulcherrima* (teleomorph *Metschnikowia pulcherrima*) (wine), *C. parapsilosis* (teleomorph *Monilia parapsilosis*) (coffee), *C. famata* (teleomorph *Debaryomyces hansenii*) (cheese), and *C. zeylanoides* (teleomorph *Kurtzmaniella zeylanoides*) and *C. norvegensis* (teleomorph *Pichia norvegensis*) (cocoa). These species are associated with the production of key metabolites (food aroma formation) and different enzymes. However, safety-associated selection criteria are often neglected. It is widely known that some *Candida* species are opportunistic human pathogens, with important clinical relevance. Here, the physiology and metabolism of *Candida* species are addressed, initially emphasizing their clinical aspects and potential pathogenicity. Then, *Candida* species used in food fermentations and their functional roles are reported. We recommended that *Candida* not be used as food cultures if safety assessments are not performed. Some safety features are highlighted to help researchers choose methods and selection criteria.

## 1. Introduction

Yeasts have been exploited by humans for millennia to produce beers, wines, and a wide range of other fermented foods. Among nonconventional yeasts, *Candida* is frequently present in cheese, coffee, cocoa, vegetables, meat, and alcoholic spontaneous fermentations [1,2,3,4,5,6]. *Candida* species are able to utilize different carbon sources and produce enzymes, acids, and other by-products [7]. Recent metagenomics approaches have also confirmed the presence of *Candida* in natural fermentations, but being rarely dominant [8]. It is because other microbial groups, including *Saccharomyces cerevisiae* and lactic acid bacteria, initiate growth and accumulate by-products (ethanol and lactic acid, respectively), inhibiting *Candida* growth.

With the advancement of large-scale fermentations, the use of starter cultures began to be developed, and numerous microorganisms were selected with specific characteristics for various food commodities. A range of studies indicate that *Saccharomyces* and *Lactobacillus* are the most widely used starter cultures in the food industry, highlighting alcoholic beverages and dairy products, respectively [9]. Nonetheless, *Candida* is also applied to produce various food commodities since its use as an adjuvant in cheese production or as starters for coffee, cocoa, vegetables, wine, beer, and meat fermentations. They are generally used in co-inoculations with *Saccharomyces* and *Lactobacillus*. For example, *C. zemplinina* (teleomorph *Starmerella bacillaris*) and *M. (Candida) pulcherrima* were used together with commercial *S. cerevisiae* to produce wines with less alcohol and more glycerol or ethyl acetate [3]. 

On the other side of the coin, *Candida* genus is well known for causing infections, especially in the gastrointestinal tract (GIT) [10]. Although most systemic infections are caused by *C. albicans*, other species have also been frequently reported in clinical studies. Some important examples include *C. utilis* (teleomorph *Cyberlindnera jadinii*), *C. tropicalis, Monilia (Candida) parapsilosis,* and *M. (Candida) pulcherrina* [3,5,6]. Therefore, the presence of *Candida* in food may represent a potential risk to public health [11]. In the plankton form, non-*Candida albicans* species (*C. krusei* (teleomorph *Pichia kudriavzevii*), *C. lusitaniae* (teleomorph *Clavispora lusitaniae*), *D. (Candida)*
*famata*, *C. colliculosa*, *M. (Candida)*
*parapsilosis*, and *C. tropicalis*) present biotypes and virulence factors similar to those of *C. albicans*, demonstrating the risk of their delivery from fermented foods [11] (Figure 1). However, Koh [12] reinforces that studies in an animal model should be carried out to prove the invasive capacity and dissemination of isolates of food origin. 

This current review summarizes the common *Candida* species used in fermented foods. Initially, we reported the physiological, metabolic, and pathogenic characteristics of the *Candida* group. Then, an association between food-related *Candida* and pathogenicity evaluation is presented for its safety in fermented products.

## 2. Physiology and Metabolism

The genus *Candida* was described for the first time in 1923 by the Danish microbiologist Christine Berkhout, even though since 1853 the species *Oidium albicans* Robin, today classified as *C. albicans*, had been known. The genus name is derived from the Latin word *candidus,* which means “white” since these microorganisms do not produce carotenoid dyes and their colonies are whitish in color [13]. About one-quarter of all yeast species belong to the genus *Candida*, which comprises around 200 species today, of which around 40 are human pathogens. Originally, *Candida* species were described as ascomycetous yeasts able to form hyphae or pseudohyphae without sexual spores. According to recent phylogenetic analyses, *Candida* species were confirmed to belong to a polyphyletic group within the *Saccharomycotina* subdivision, under the *Ascomycota* division, *Hemiascomycetes* class, *Saccharomycetales* order, and *Saccharomycetaceae* family in the fungal kingdom [13,14,15].

*Candida* is classified as an anamorphic genus because of the lack of a sexual reproduction stage. Asexual multiplication (mitosis) in *Candida* is characterized as multipolar or multilateral, where budding occurs at many sites on the cell wall. However, several *Candida* species have now been shown to undergo mating and meiosis, and population studies indicate that, for some species, sexual reproduction is at least a possibility [16]. The sexual reproductive stage, denominated teleomorph, can be differentiated from the corresponding anamorph by phylogenetic analysis [17], in the sense that a species may have two names that correlate to the growth stage of the yeast. Examples are *Candida pelliculosa* (anamorph) × *Wickerhamomyces anomalus* (teleomorph); *Candida guilliermondii* (anamorph) × *Meyerozyma guilliermondii* (teleomorph); *Candida kefyr* (anamorph) × *Kluyveromyces marxianus* (teleomorph); *Candida krusei* (anamorph) × *Pichia kudriavzevii* (teleomorph); *Candida lipolytica* (anamorph) × *Yarrowia lipolytica* (teleomorph); *Candida lambica* (anamorph) × *Pichia fermentans* (teleomorph); *Candida utilis* (anamorph) × *Cyberlindnera jadinii* (teleomorph) (CDC 2021).

Yeast cells of the genus *Candida* are usually oval, ellipsoidal, or elongated, with sizes in the range of (1–8) × (1–6) μm^2^. They can grow as single cells (cell budding with separation of the newborn cells), as pseudohyphae (cell budding without separation of the newborn cells), or as true hyphae (no budding, hyphae formation with transverse walls). Yeasts that reproduce by hyphae formation in addition to budding are classified as dimorphs and are represented by the species *C. albicans*, *Candida amapae* (teleomorph *Saccharomycopsis amapae*), *Candida bertae*, *Candida castrensis*, *Candida chilensis*, *Candida chiropterorum*, *Candida edax*, *Candida entomophila*, *Candida fennica*, *Candida homilentoma* (r teleomorph *Hyphopichia homilentoma*), *Candida insectorum*, *Candida lipolytica* (teleomorph *Yarrowia lipolytica*), *Candida lyxosophila*, *Candida mesenterica*, *Candida paludigena*, *C. railenensis*, and *Candida tropicalis* [13,14].

*Candida* is a heterogeneous genus that comprises microorganisms with different physiologic and metabolic characteristics. The guanine-cytosine content of the DNA can vary between 30 and 62.7%, and there are four ubiquinone types within the genus, namely, CoQ 6, 7, 8, and 9. Regarding the use of carbon sources by the strains already described, 71% of *Candida* species utilize xylose, 57% utilize cellobiose, 29% are able to oxidize alkanes, 27% can utilize starch, and 7% can use methanol. This nutrition profile is in accordance with the fact that most *Candida* species are associated with plants, decaying biomass, insects, and foods. Besides, about 85% of the species are dependent on vitamins synthesized by plants. Concerning their growth temperature, most *Candida* species are mesophilic. They grow well in temperatures between 20 °C and 25 °C, but few species are able to grow above 37–40 °C [14]. Optimal growth temperatures range between 25 °C and 30 °C, while optimal pH values are between 4.0 and 6.0. *Candida* strains are usually aerobic and can produce small amounts of alcohol during aeration. Many strains synthesize other extracellular metabolites, such as citric acid, xylitol, erythritol, biosurfactants, exopolysaccharides, and enzymes [13].

Different species of *Candida* are known to produce sophorolipids—extracellular glycolipids that are the most common biosurfactants used in cosmetics and pharmacology. One example is the mannosyl erythritol lipid, a yeast glycolipid produced from vegetable oils by *Candida* strains [18]. A novel biosurfactant with antibacterial activity against pathogenic *Escherichia coli* and *Staphylococcus aureus* was obtained from *M. (Candida)*
*parapsilosis*. The biosurfactant was identified as a docosenamide-type compound with a 337.5 g/mol [19].

*P. (Candida) krusei* is an acid-resistant and thermo-tolerant species, able to grow at low pH (3–4) and high temperatures (40–43 °C). The proposition that *P. (Candida) krusei* is the anamorph of *P. kudriavzevii* dates from 1980, by Kurtzman and collaborators. This strain can also be identified in the literature as *Issatchenkia orientalis* and *Candida glycerinogenes*. The yeast is naturally present in milk and in many other fermented foods such as cheese, sourdough breads, kefir, fermented cassava, and cocoa fermentation. It does not excrete any extracellular toxins and can be used, for example, in the synthesis of single-cell proteins [20]. This strain is employed for industrial-scale production of glycerol and succinate and is also able to synthesize bioethanol [21].

Some *Candida* species, such as *C. pelliculosa*, *M. (Candida)*
*parapsilosis*, and *Candida pyralidae*, demonstrate killer activity against undesired or pathogenic organisms. This activity, first described by Bevan and Makower in 1963 on an *S. cerevisiae* strain, is attributed to the production of killer toxins or killer factors, which are proteinaceous antimicrobial compounds secreted by yeasts. The killer activity of *Candida* yeasts can be useful to control spoilage organisms in various environments, such as fermented foods and beverages [22,23]. 

## 3. Clinical Relevance 

Most *Candida* species live in commensalism relation with animals and humans, being part of the resident or transient gut microbiota, mucocutaneous tissues, and skin. Studies indicate that 70% of the healthy human population carries, asymptotically, *Candida* species [24]. However, among the more than 200 *Candida* species reported, about 20% are considered pathogenic. *C. albicans*, the most prevalent and invasive species, are frequently isolated from the nosocomial environment and GIT. This species is one of the most representative of what is understood as an opportunistic pathogen, given its prevalence in healthy individuals. At the same time, it consists of the etiological agent most commonly related to systemic fungal infections [25]. 

Contamination by *Candida* sp. was considered superficial and rare until the early 1960s [26]. Since then, new cases have become frequent in hospital infections. It is speculated that the widespread use of antibiotics was the indirect cause of the increase and dissemination of contamination by *Candida* sp. The hypothesis is that, by eliminating bacteria from the human microbiota through antibiotics, *Candida* and other yeasts have increased their numbers mainly in the GIT. Furthermore, antitumor drugs can potentially damage the mucosa of the GIT, facilitating invasion through the membrane of enterocytes into the bloodstream and, consequently, the spread of the fungus to the other organs [11].

The most frequent reservoirs of *C. albicans* are superficial infections affecting the skin, mucous membranes of the female genital system, oral cavity, and GTI. However, in healthy individuals, yeast is in significantly lower populations than other microbial groups, since only 0.1% of the human microbiome corresponds to eukaryotic microorganisms or viruses [27]. An increase in yeast numbers can occur through a change in the microbial population in the mucous membranes with a preponderant growth of *Candida*.

In immunocompromised patients, the presence of *Candida* sp. in the bloodstream results in a wide range of severe manifestations. This condition, known as candidemia, affects different organs, such as the liver, spleen, and eyes [28,29]. The main mechanisms involved in the pathogenesis of invasive candidiasis include the following: imbalance of the gut microbiota, which enables the significant growth of *Candida* species in the intestine; some lesions to the GIT mucosa, which favor invasion by the fungus; impairment of the host’s immune response, which may result in severe generalized infection [24].

As mentioned, *C. albicans* can be present in the form of unicellular yeast, multicellular pseudohyphae, or filamentous hyphae. The morphogenesis from leveduriform to a hypha state is an important virulence mechanism [30]. In general, the leveduriform state is more associated with the relationship of commensalism with the host, and hyphae form with the condition of pathogenicity (Figure 1). One of the reasons for this phenomenon is that hyphae yeasts are less likely to be phagocytized, being able to escape from the host defense cells [31]. However, in non-*C. albicans* (NCAC) species, the transition from the leveduriform form to hyphae is not a condition for invasion of host tissue, as observed for *Candida glabrata* (teleomorph *Nakaseomyces glabrataa*) and *M. (Candida)*
*parapsilosis* [32]. In any event, the balance between both states depends on the components of the fungus’s cell wall, known as pathogen-associated molecular patterns (PAMPs), represented primarily by human host cell pattern recognition receptors (PRRs) [33].

*Candida* harbors a wide variety of ecological habitats in nature due to its versatility in acquiring nutrients, metabolic flexibility, and resistance to stress factors [25]. In GIT, the transition from commensal to pathogenic form involves the adaptation to changes in pH, oxygen, and nutrient availability, and the interaction with the rich and diverse resident microbiota. Subsequently, different mechanisms favor the adhering and invasion of host cells; once disseminated, *C. albicans* cells must protect themselves and adapt to the host’s immune system, which involves different signaling factors and networks [25].

*C. albicans* corresponds to the main antifungal T helper 17 (Th17) cell inducer in humans [34]. Such cells correspond to key pieces in the body’s immune responses against the invasion of bacteria and fungi, organizing barrier immunity, especially on the skin and mucosa of GIT [35]. Immune responses related to specific T cells of *C. albicans* broadly modulate antifungal immunity through cross-reactions with other species, constituting a central mechanism for systemic induction of antifungal responses in humans as well as a risk factor for inflammatory lung diseases caused by other fungal species [34]. Unbalanced immune responses, dependent on Th17 cells, may also contribute to the development of different types of disorders, including inflammatory bowel diseases [36]. In addition, the role of the mycobiome and its influence on specific conditions, such as autism, metabolic syndromes, chronic viral infections, and even tumorigenesis, has been investigated [33,37,38].

The virulence factors of *C. albicans* and NCAC are fundamental for colonization and invasiveness. However, the efficiency of colonization varies among individuals, being affected by immunological competence, use of medications with consequent impairment of the gut microbiota, and oral hygiene, among others that may result in fungal dysbiosis (i.e., intestinal flora imbalance) [30,39,40]. The main virulence factors include (i) growth at 37 °C, (ii) secretion of hydrolytic enzymes (which confer the capacity to destroy host tissues), (iii) epithelial adhering ability (essential for surface colonization), (iv) hemolytic activity (which enables the degradation of red blood cells and consequent access to nutrients), and (v) formation of biofilm (important for the colonization of surfaces, as well as for resistance to antifungal agents) [11]. The latter is significant for *Candida* of food sources, given the ability to colonize surfaces that encounter food in the production environment and packaging.

Although *C. albicans* is commonly used as a model organism for fungal pathogens, the relevance of other species has also been demonstrated, such *as N. (Candida) glabrata*, *M. (Candida)*
*parapsilosis*, *C. tropicalis*, *P. (Candida) krusei*, and *Candida guilliermondii* (r teleomorph *Meyerozyma guilliermondii*) [41]. Hirayama et al. [42] evaluated the degree of pathogenicity of six important virulent species of *Candida*. Mice fed a low protein diet were inoculated intragastrically with *Candida* cells, in addition to receiving oral antibiotics and cyclophosphamide (an immunosuppressive drug) to facilitate fungal invasion and dissemination in the GTI. *C. albicans* and *C. tropicalis* resulted in higher mortality rates, and the former demonstrated higher invasiveness power, resulting in marked necrosis in liver tissues.

The condition of mycobiome eubiosis—that is, balanced interaction between host and fungi [43]—still needs further elucidation to understand in what proportions, in the GTI, *Candida* is classified as low risk for humans. It may vary depending on factors such as gender, age, type of diet, and associated comorbidities [38]. The ingestion of probiotics has been suggested as an important strategy to prevent the dysbiosis caused by *Candida* [44]. *Lactobacillus* and *Bifidobacteria*, for example, produce organic acids from carbohydrate intake, reducing the GTI pH and, consequently, inhibiting the growth of *Candida* and other pathogens [45]. Some additional antagonism factors include competing for nutrients and the production of toxic molecules (e.g., bacteriocins, biosurfactants, and hydrogen peroxide) [46]. Furthermore, the fact that probiotics affect the development of *Enterobacteriaceae* can prevent intestinal colonization by *C. albicans*; members of the Enterobacteriaceae family produce liposaccharides that induce *Candida* growth in the GTI [30,46].

## 4. Isolation and Cultivability 

The correct and rapid etiological identification of *Candida* from complex samples is usually performed by MALDI-TOF mass spectrometry (MS) and Vitek card [47]. However, these methods can result in the misidentification of some strains. *Candida auris*, for example, are mistakenly identified by Vitek 2^®^ (bioMérieux, Marcy-l’Étoile, France) as *C. haemulonii*, *C. duobushaemulonii*, *D. (Candida)*
*famata*, or *N. (Candida)*
*glabrata* [48]. Thus, the identification of *Candida* species by conventional morphology and assimilation tests is still being used today. The conventional medium mostly used to isolate *Candida* spp. is Sabouraud glucose or dextrose agar (SDA). However, SDA supports general pathogenic fungal requirements and does not guarantee the differentiation between species in mixed yeast cultures [49]. Usually, SDA is incubated aerobically at 37 °C for 24–48 h, and grown colonies show a cream, smooth, pasty convex pattern [50]. The diversity of isolation media and incubation conditions are shown in Table 1. Another example of a conventional cultivation medium is Pagano-Levin, which contains triphenyltetrazolium chloride in its composition, and, when reduced, it can distinguish *C. albicans* from other species due to its pale colonies alongside pinkish non-*C. albicans* [50]. Pagano-Levin agar has a similar sensitivity to SDA but is superior for the detection of more than one species in the sample. 

Subsequently, improved culture media arose to allow, besides cultivation, species differentiation, and presumptive identification in complex samples. Chromogenic formulation agars, such as CHROMagar Candida (CAC) and Candida ID, have high sensitivity and specificity for more than 99% of isolates [75]. The species can also be identified 24 h earlier than with conventional culture media [52]. Candida ID agar is based on the chromogenic indolyl glucosaminide substrate, which is hydrolyzed by *C. albicans* and *C. dubliniensis* to yield a blue product. The CAC medium contains chromogenic beta-glucosaminidase substrate that reacts with species-specific enzymes to give colonies with different colors [49]. This medium allows the identification of *C. albicans*, *C. tropicalis,* and *P. (Candida) krusei*. Some reports have suggested that CAC can provide an indication of the presence of *C. albicans, C. tropicalis,*
*P. (Candida) krusei*, *N. (Candida)*
*glabrata*, and discrimination of *C. dubliniensis* from *C. albicans* [50,75]. Colorful indication represents green to *C. albicans*, dark green to *C. dubliniensis*, blue to *C. tropicalis*, pale to *P. (Candida) krusei*, and pink to *C. auris* [50]. Using this medium, the temperature of incubation seems to be an additional differentiator between species, as well as carbon source and salinity. Incubation at 45 °C, for example, failure to grow *C.*
*dubliniensis*, while promoting *C. albicans* growth [90]. Giammanco et al. [76] failed to grow *D. (Candida) pararugosa*, *D. (Candida) rugosa*, *C. maris*, *C. silvanorum* and *Candida magnoliae* (teleomorph *Starmerella magnoliae*) in SBA and CAC at 37 °C. *N. (Candida)*
*glabrata* showed restriction to the use of dextrose as the only carbon source, and *C. auris* was the only isolate that grew at high salinity (10% wt/vol) on SAB in comparation to other species [54]. To overcome these failures, some modified culture media are proposed to increase isolation efficiency. Examples include SCA medium that adds crystal violet (0.5 mg/L) to differentiate *C. auris* from *C. tropicalis* [91], the addition of sodium chloride and ferrous sulfate in yeast extract-peptone-dextrose (YPD) agar named Selective Auris Medium (SAM) to isolate *C auris* [48], and CAC with Pal’s medium addition to improve differentiation of *C. dubliniensis* from *C. albicans* [92].

*Candida* species are also isolated from animals of food importance, such as cows, avians, sows, and geese. *C. albicans*, *N. (Candida)*
*glabrata*, *C tropicalis*, *C. (Candida) utilis,* and *M. (Candida)*
*parapsilosis* are the main reported species [46,77,84] isolated from SBA and chromogenic agar. These cultivation media are useful for monitoring food source animals, which are considered the most important vehicle in transmitting pathogenic food-borne microorganisms to humans [93]. The World Health Organization (WHO) states that illness due to contaminated food is perhaps the most widespread health problem and an important cause of reduced economic productivity [94]. The predominance of the *Candida* genus and the presence of *C. albicans* were reported in various cattle mastitis cases contaminating milk during udder, affecting milk quality in the dairy industry and consumers’ safety [79]. Another cross-infection *C. albicans* route can be through raw meat, e.g., chicken and mutton [46]. This suggests the need to control hygienic parameters for the handling and storage of raw meat [77]. Differently from clinical and food sources, studies of *Candida* associated with animal sources use more generalist culture media, such as potato dextrose agar (PDA), yeast malt agar (YM), and yeast peptone glucose agar (YPD), to increase the isolation spectrum. Usually, it is supplemented with chloramphenicol to avoid bacterial growth [87,88,89]. 

## 5. *Candida* Diversity in Cultured Foods and Biotechnological Importance 

*Candida* species are common microorganisms identified in the phyllosphere (e.g., leaves, flowers, fruits, and stems), playing pivotal roles in ecological interactions between superior plants-insects and agricultural practices [95,96]. These epiphytic residents are able to consume plant nutrients and thrive in the interior of plant tissues; their prevalence is associated with edaphoclimatic conditions, geographical location, plant genotype, and fruit ripening stage [97,98,99]. A study conducted by Vadkertiová et al. [96] evaluated the diversity and prevalence of yeasts in the blossoms and fruits of apple, pear, and plum trees. The results indicated a higher overall diversity observed in blossom tissues, which can be associated with cross-contamination of commensal yeasts present in the GTI of insects [100]. Regarding *Candida*, *C. tropicalis* was the only species identified in both plum blossom and fruit, while *Candida boidinii* and *Candida catenulata* were present exclusively in the blossom of apple and pear, respectively. Jones et al. [101], on the other hand, investigated the relative abundance of yeasts in different maturation stages of blueberries, cherries, raspberries, and strawberries. The quantitative analysis using the metabarcoding approach revealed that, although *Candida* species represented only 2.4% of the total OTU reads, they were the most diverse genus, accounting for over 20% of the identified phylotypes. *S. (Candida) zemplinina*, *C. rancensis*, *C. (Candida)*
*membranifaciens* (teleomorph *Candida flareri*), *C. argentea*, *C. railenensis*, and *C. kofuensis* were the most relevant species reported. 

These evidences reinforce the importance of *Candida* species to the food industry since this group of yeasts is part of the wild microbiota of many fermented foods. A recent study by Arroyo-López [102] evaluated *Candida* diversity during fruit reception, brine, and table olive fermentation. *C. diddensiae* and *M. (Candida) parapsilosis* were identified during all processing stages, while *C. tartivorans* and *C. tropicalis* were restricted to brine and fermented fruits, respectively. Interesting results were also evidenced by Abdo et al. [103], which evaluated the prevalence and diversity of fungi species on floors, walls, and equipment before and up to two years after the arrival of the industrial plant. Using a metabarcoding approach, the study evidenced that the *Candida* genus was found only in used machinery before the first harvest, whereas showing an incremental relative prevalence after subsequent harvests. However, this pattern is not restricted to the previously mentioned substrates. After the analysis of the metadata from over 120 published papers, Pereira et al. [9] stated that *Candida* was the only genus reported in all the different types of fermentation (alkaline, alcoholic, acetic, lactic, and mixed processes) associated with numerous fermented foods and beverages.

Apart from its ubiquitous presence, *Candida* also possesses a complex metabolic machinery that allows its survival, competition, and, sometimes, dominance during fermentative processes [104]. Some species (e.g., *Candida versatilis* (teleomorph *Wickerhamiella versatilis*), *S. (Candida) zemplinina*, and *S. (Candida) magnoliae*) are osmotolerant and able to grow in musts with increased sugar content or stressful conditions due to high salt concentrations [105,106,107]. The presence of these stressful conditions increases glucose uptake due to changes in the affinity of membrane transporters encoded by *SNF3*, *HXT1*, and *HXT2* genes and the redirection of the glycolytic flux towards glycerol production [108,109]. *S. (Candida) zemplinina*, frequently isolated from oenological *terroir* [110], also exhibits a fructophilic nature (i.e., preference of fructose over glucose as the main carbon source), a feature recently explored in fermented products with high residual sugar content, such as cocoa and coffee [111,112]. Last but not least, several *Candida* species can also produce a wide range of flavor and odor-active organic compounds, such as esters, aldehydes, and higher alcohols, through assimilation and metabolization of amino acids and higher alcohols via the Ehrlich pathway and esterification reactions, respectively [113,114]. 

All these factors have increased the interest in research that uses *Candida* as starter cultures for fermented foods. Table 2 shows some examples of *Candida* species and their applications in fermented foods. Due to the driven evolution caused by global warming, grape musts are showing superior sugar content, which may lead to an undesirable increase in the ethanolic grade of wines produced by pure *Saccharomyces* fermentation [115]. In this sense, several researches are exploiting the potential of non-*Saccharomyces* yeasts for mixed and sequential fermentation processes [116,117]. The inoculation of pure *S. (Candida) zemplinina* achieved a similar cell count compared to *S. cerevisiae*. However, an inhibitory effect against *S. cerevisiae* could be observed when *S. (Candida) zemplinina* was inoculated previously [1]. These observations can be associated with the yeast-specific nitrogen assimilation profiles, where histidine, methionine, threonine, and tyrosine are preferred consumed by *S. (Candida) zemplinina* in temperatures of up to 28 °C, resulting in poorer performances of *S. cerevisiae* [118]. Apart from ecological interactions between the starter cultures, the addition of *S. (Candida) zemplinina* resulted in a superior aroma complexity due to enhanced production of esters, higher alcohols, and free terpenes [1,119,120].

Mixed fermentations, such as cocoa and coffee beans, harbor a diverse and heterologous group of microorganisms, which directly contribute to the aroma and flavor development of the final products. *M. (Candida) parapsilosis* have been selected based on its high pectinolytic activity, resistance to stressful conditions (high temperatures, acidity, high ethanol concentration, among others), killer factor, and production of aroma-active compounds [5,131]. The pure inoculation of *M. (Candida) parapsilosis* in cocoa and coffee fermentation resulted in significant changes in sugar and amino acids inside the beans, which are important precursors for the aroma development [5,122,124]. However, these traits resulted in a higher residual sugar content in the fermenting pulp-bean mass, which could lead to the development of spoilage microorganisms during the drying process.

The use of *Candida* species as starter cultures in fermented milk has also been exploited. Biagiotti et al. [127] investigated the yeast microbiota involved in the ripening of Italian Fossa cheese, revealing the presence of *K.* (*Candida*) *zeylanoides* and *H. (Candida) homilentoma* during the process. To assess the impact on the ripening process, the inoculation of *C. zeylanoides* and *H. (Candida) homilentoma* resulted in lower mold colonization and desirable sensorial characteristics, including flavor quality, intensity, and body texture. Besides technological and sensorial aspects, the antioxidant and antimicrobial activity against *Bacillus subtilis* has also been explored in bovine colostrum fermentation after *Y. (Candida)*
*lipolytica* inoculation, which demonstrated superior results in comparison to a commercial probiotic culture [126]. 

## 6. *Candida* Pathogenicity Evaluation

Different *Candida* strains used as starter cultures, including *P. (Candida) krusei*, *M. (Candida)*
*parapsilosis*, *M. (Candida)*
*guilliermondii*, and *K. (Candida)*
*kefyr*, have been associated with some type of fungal infection, although they are generally considered less virulent than *C. albicans* [91,132,133]. Their classification can be controversial between nonpathogenic yeast or non-albicans opportunistic pathogenic yeast. Thus, the pathogenicity and drug resistance of *Candida* strains must be previously evaluated before their use in foods and beverages. Virulence attributes can be evaluated by testing adhesion ability, biofilm formation capacity, production of phospholipases and proteinases, dimorphic transition, susceptibility to antifungal agents, and immune host response. 

Firstly, *Candida* starter candidates can be evaluated for their genetic similarity with recognized pathogenic yeasts (e.g., *C. albicans*) as a primary assessment. Phylogenetic tree analysis, through the neighbor-joining method, revealed proximity between *C. tropicalis* and *C. albicans*, for example [7]. Moreover, *P. (Candida) krusei* was considered an opportunistic pathogenic yeast and is closely related to the well-defined opportunistic *N. (Candida)*
*glabrata*. Secondly, adherence ability to host epithelial cells must be accessed, which is the first step to colonization and invasion. Key virulence genes (e.g., Hwp1p, EAP1, CaMnt1p, and Csf4) can be checked by the sequencing method or specific PCR primer sets. Moreover, epithelial adherence can be tested by an adhesion assay. Segal et al. [134] showed that buccal epithelial cells were incubated with *Candida* cells for 2 h. Biofilm formation can also be quantified. Jin et al. [135] used microtiter plates to produce *Candida* biofilms while cell suspension was aspirated and washed to remove loosely adherent cells. Then, cells were quantified by 2,3-bis(2-methoxy-4-nitro-5-sulfophenyl)-5-[(phenylamino) carbonyl]-2H-tetrazolium hydroxide (XTT) reduction and crystal violet assay. 

Hydrolytic enzymes (e.g., phospholipases and proteinases) can be secreted by pathogenic fungi and estimate invasiveness. Secretion is linked to the presence of aspartyl proteinase (SAP1–SAP8) and phospholipase (PLB1 and PLB2) genes [136]. Extracellular phospholipase activity can also be evaluated by the plate method. The presence of halo means precipitation from hydrolysis of phosphatidylcholine by concentrated culture filtrates [137]. To access the strain drug resistance, an in vitro evaluation of antifungal susceptibility can be performed via a minimum inhibitory concentration (MIC) assay using innumerous antimycotics, such as azoles (fluconazole (FL), ketoconazole (KE), and itraconazole (IT)), amphotericin B (APB; polyenic) and flucytosine (FC; pyrimidine) on Petri dishes [138].

More meticulous tests can be performed, including yeast morphological modifications and host immunological response. Some *Candida* species present dimorphic transitions, enabling switching between yeast and hyphae morphologies. Yeast shape is usually non-invasive, while the hypha apparatus is prone to penetrating human cells [139]. Yang et al. [140] observed by microscopy the hyphae development of *C. albicans* in both liquid and agar media (YPD+10% fetal bovine serum). The authors also evaluated the interaction between yeast and oral epithelial cells and performed endocytosis, invasion, and damage in in vitro assays. The intensity of *Candida* infection depends directly on the host immune system response. Studies have shown that *N. (Candida)*
*glabrata* and *C. albicans* infection induce the specific proinflammatory cytokines TNF-a, IL-12, and IFN-g, being TNF-a the major one in host defense against systemic candidiasis cases. In vivo evaluation of pathogenesis was analyzed by Brieland et al. [141]. Mice kidney samples were submitted to real-time RT-PCR to study the kinetics of cytokine induction and to ELISA kits to analyze the immunoreactivity of cytokines. 

## 7. Conclusions

The use of *Candida* as starter cultures has shown important roles in fermented foods. They increase the content of aromatic compounds in wines and fermented milks and contribute to the pulping of cocoa and coffee beans, in addition to participating as an adjunct in the production of cheeses. However, there are serious concerns about infections caused by this yeast group in humans and animals. Although most cases of invasive yeast infection are attributed to *C. albicans*, there are increasing rates of non-*C. albicans* species in various parts of the world. Therefore, we recommend that pathogenicity tests be performed as a precondition for the use of *Candida* in food studies. These tests include the production of molecules that mediate adhesion to and invasion into host cells, the secretion of hydrolases, the yeast-to-hypha transition, biofilm formation, phenotypic switching, and a range of fitness attributes. Furthermore, safety-related genes, including harmful metabolite production and adhesion ability, can be studied using next-generation sequencing techniques or other mechanistic omics tools. Nonetheless, genomics may reveal only a theoretical risk level because gene expression is dependent on environmental conditions. Thus, genomic data must be confirmed by in vitro studies and human clinical trials for the safe use of *Candida* in foods.

## Figures and Tables

**Figure 1 microorganisms-10-01855-f001:**
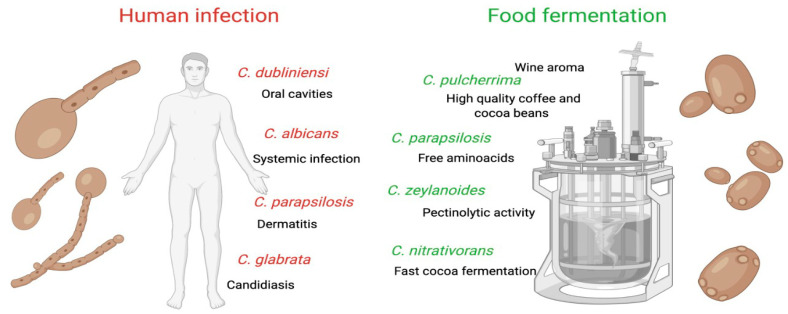
*Candida* species dual effect.

**Table 1 microorganisms-10-01855-t001:** *Candida* species isolation mediums and conditions.

Species	Isolation Source	Medium	Cultivation Conditions	References
Clinical Samples				
*C. dubliniensis*	Denture wearers	SAB and CAC	Room temperature	[51]
*C. albicans*, *N. (Candida)**glabrata*, *P. (Candida)**krusei C. tropicalis*, *M. (Candida)**parapsilosis*, *C. (Candida) lusitaniae*, *M. (Candida) guilliermondii*, *D. (Candida)**famata*, *C. lambica*, *K.* (*Candida*) *zeylanoides*, and *Candida humicola* (teleomorph *Vanrija humicola*)	Clinical samples	SAB and CAC	48 h, 37 °C	[52]
*Candida* spp.	Human oral cavities	SAB with chloramphenicol;	NI	[53]
*C. auris*	Clinical samples	CAC	40 °C	[54]
*Candida* spp.	Oral cavity	SDA and CAC	24–48 h, 37 °C	[50]
*C. dubliniensis*	Denture stomatitis	Candida ID2	48 h,37 °C	[51]
*Candida* spp.	Blood sample	CAC	30 °C	[55]
*C. albicans*, *C. dubliniensis*, *C. tropicalis*, and *P. (Candida)**krusei*	Clinical samples	Candida ID, Albicans ID 2, CAC	10 days, 37 °C	[56]
*Candida* spp.	Oral cavity	SAB	48 h, 37 °C	[57]
*C. africana*	Vagina sample	CAC	NI	[58]
*C. dubliniensis*	Oral cavity	CAC	48 h, 30 °C	[59]
*C. africana*	Oral candidiasis sample	SAB	24 h, 30 °C	[60]
*C. rugosa* (teleomorph *Diutina rugosa*) and *C. mesorugosa* (teleomorph *Diutina mesorugosa*)	Clinical samples	SAB and brilliance Candida agar	NI	[61]
*N. (Candida)* *glabrata*	Blood samples	Blood agar, chocolate agar, SAB, and MH agar	NI	[62]
*C. albicans*, *P. (Candida)**krusei*, *C. tropicalis*, and *N. (Candida)**glabrata*	Urine	CAC, McConkey and blood agar	48 h, 30–37 °C	[63]
*Candida* spp.	Vagina	SAB and Hichrome Candida differential agar	24–48 h, 37 °C	[64]
*M. (Candida)* *parapsilosis*	Scalp	SAB with streptomycin, SAB, and BHI with streptomycin and penicillin	3 days, 37 °C/room temperature	[65]
*Candida* spp.	Clinical samples	SAB with chloramphenicol	72 h, 28 °C	[66]
*C. albicans*, *N. (Candida)**glabrata*, *C. dubliniensis*, *P. (Candida)**krusei*, and *M. (Candida) parapsilosi*	Oral cavity	CAC	3–4 days, 37 °C	[67]
*C. albicans*, *C. tropicalis*, *P. (Candida)**krusei*, *N. (Candida)**glabrata*, and *C. dubliniensis*	Clinical specimens	CAC	24 h, 37 °C	[68]
*C. africana*	Vagina	CHROMagar and Candida ID2	NI	[69]
*Candida* spp.	Denture-related stomatitis patients	SAB	7 days, 35 °C	[70]
*C. auris*	Healthy human ear	CAC, Candida ID, SAB, and malt extract agar	48 h, 37 °C	[71]
*D. (Candida)* *famata*	Eye infection	YEPD	30 °C	[72]
*C. albicans, N. (Candida)**glabrata* and *C. kefyr* (teleomorph *Kluyveromyces marxianus)*	Vulvovaginal candidiasis patients	SAB with chloramphenicol and cycloheximide	48 h, 30 °C	[73]
*N. (Candida)* *glabrata*	Blood	CAC, SAB IMA, and MYC	48 h/4–7 days, 30–37 °C	[74]
*C. albicans*, *C. tropicalis* and *P. (Candida)* *krusei*	Blood	CAC	48 h, 37 °C	[75]
*C. pararugosa* (teleomorph *Diutina pararugosa*)	Oral cavity	SAB with chloramphenicol and CAC	37/25 °C	[76]
**Food Samples**				
*M. (Candida)* *parapsilosis*	Contaminated dairy products	PDB	24 h, 37 °C	[19]
*C. albicans*	Raw chicken meat and mutton meat	SAB	7–10 days, 30 °C	[77]
*Candida* spp.	Mastitic bovine milk	Blood agar, Mac Conkey agar, and SAB	72 h, 37 °C	[78]
*C. albicans*	Landes geese	CAC, YPD, TTC-Sha Baoluo, Corn meat agar medium	72 h, 35 °C	[46]
*C. albicans*	Milk	SAB with chloramphenicol and CAC	24 h, 37 °C	[79]
*Candida* spp.	Mastitic cows and milkers	SAB	72 h, 37 °C	[80]
*C. tropicalis*	Infected sows	SAB and TSA with newborn bovine serum	3–5 days, 28/37 °C, respectively	[81]
*N. (Candida)* *glabrata*	Avian species	SAB	48 h, 37 °C	[82]
*D. (Candida) rugosa*	Turkeys	SAB	2 weeks, 27 °C	[83]
**Environmental Samples**				
*C. auris* and *N. (Candida)* *glabrata*	Plastic health care surface	SAB; YNB	40 °C	[54]
*C.* (*Candida*) *utilis*	Hulu Mur, Kissra, and Marisa (Sudanese traditionally cereal-based fermented foods)	YM with rose bengal	3 days, 28 °C	[84]
*C. auris*	Sediment soil and seawater Coastal wetlands (e.g., rocky shores, sandy beaches, tidal marshes, and mangrove swamps)	SAB with chloramphenicol and gentamicin, CAC	7 days, 28 °C	[85]
*C. albicans*	Fresh water and sewage	Ágar MacConkey	24 h, 22 °C	[86]
*C. albicans* and *Y. (Candida) lipolytica*	Soil samples	PDA with chloramphenicol, CAC	3–5 days, 28 ± 2 °C	[87]
*C. digboiensis*	Acid mine drainage	GPYE	32 ± 2 °C	[88]
*C. (Candida) membranifaciens subsp. flavinogenie strain W14-3*	Seawater and sediments	YPD with chloramphenicol	5 days, 20–25 °C	[89]

NI-not informed; SAB-sabouraud dextrose; CAC-CHROMagar Candida; MH-Mueller Hinton; YEPD-yeast extract peptone dextrose; IMA-inhibitory mold agar; MYC-Mycosel; PDB-potato dextrose broth; PDA-potato dextrose agar; YPD-yeast peptone glucose; TSA-tryptic soy agar; YNB-yeast nitrogen base; YM agar-extract-malt extract agar; GPYE-glucose peptone yeast extract.

**Table 2 microorganisms-10-01855-t002:** Selected *Candida* starter cultures, isolation source, and impacts over the sensorial and functional aspects of economically relevant fermented foods and beverages.

Fermented Product	Starter Culture	Isolation Source	Desirable Traits/ Inoculation Effects	References
Wine	*S. (Candida) zemplinina*	Grapefruits and musts	Dominance observed during sequential inoculation with *S. cerevisiae.* Higher aroma complexity during co-inoculation (↑↑ production of aliphatic alcohols and free terpenes);	[1]
		Organic wine must	Synergism when co-inoculated with *S. cerevisiae*. ↑↑ quantitative and qualitative content of esters	[119]
		Spontaneous wine fermentation	↓↓ concentration of residual sugar at the end of fermentation. ↑↑ glycerol production on trials containing the *S. (Candida) zemplinina*; Improvement of fruity and floral sensory notes when co-inoculated with *S. cerevisiae* and *Lactiplantibacillus plantarum*	[120]
	*C. railenensis*	Vineyards	Decrease in ethyl and acetate esters production. Higher production of glycerol, acetate, free terpenes, and norisoprenoids	[2]
	*W. (Candida) pulcherrina*	Spontaneous wine fermentation	Slow rate of sugar consumption. ↑ production of ethyl acetate in comparison to pure *S. cerevisae* wine fermentation	[3]
Cocoa	*C. nitrativorans*	Spontaneous cocoa beans fermentation	Natural adaptation to the cocoa fermentative process. Superior sugar metabolism. Killer phenotype trait. ↑↑ polygalacturonic and pectin lyase production	[4]
	*C. parapsilosis*		Uniformity of well-fermented seeds (>95%); Lower acidity of the cotyledon in comparison to other inoculums. ↑↑ residual sugar at the end of fermentation. ↑ free amino acids content inside the cotyledon	[5]
	*Candida* sp.	NI	Growth inhibition of pathogenic microorganisms. No significant alterations on cocoa quality parameters	[121]
Coffee	*M. (Candida) parapsilosis*	Coffee fruits	↑ proteolytic activity during semi-dry processing. Final beverage with prevalence of caramel- and fruity-like flavor, and dense body. Highest number of aroma-active compounds in semi-dry processing	[122,123,124]
Kefir	*K. (Candida) kefyr*	NI	↑↑ proteolytic activity. Decrease on total fat content. ↑↑ residual sugar at the end of fermentation;	[125]
Bovine colostrum	*Y. (Candida) lipolytica*	NI	Increased antioxidant and antibacterial activity. Inhibition of *Aspergillus niger* sporulation	[126]
Cheese	*K.* (*Candida*) *zeylanoides* and *H. (Candida). homilentoma*	Spontaneous Italian Fossa cheese	Lower contamination by molds. ↑↑ sensorial attributes in comparison to spontaneous fermentation	[127]
Table olive	*C. boidinii*	Leucocarpa cv. olives brine	Alteration of color parameters. ↑↑ enzymatic activity, resulting in the texture loss of the fermented olives. ↓↓ acidification of the brine and presence of spoilage microorganisms at the end of the fermentation	[128]
Meat	*C. deformans* and *K.* (*Candida*) *zeylanoides*	Native dry-cured *lacón*	Lower production of flavor- and odor-active compounds associated to dry-cured meat	[129]
Beer	*C. tropicalis*	Traditional sub-Saharan sorghum beer	↑↑ growth rate and cell concentration in comparison to *S. cerevisiae*. Lower glucose consumption rate. ↑ production of lactic acid in the final product. High viability rate in superior concentrations of ethanol	[6]
Soy sauce-like condiment	*W. (Candida) versatilis*	NI	Superior osmotic tolerance and growth rate at 10% NaCl soy whey medium. Superior acidification of the final product. ↑↑ residual sugar at the end of fermentation. ↑↑ contribution to umami taste Production of volatile compounds associated with desirable flavor in soy sauce	[130]

NI-not informed; ↑: higher; ↑↑: much higher; ↓↓: much lower.

## Data Availability

This study did not report any data.

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
