# Peer review of "What Is Candida Doing in My Food? A Review and Safety Alert on Its Use as Starter Cultures in Fermented Foods"

_microorganisms, 2022, doi:10.3390/microorganisms10091855_

Round 1

Reviewer 1 Report

The manuscript "What is Candida doing in my food? A review and safety alert on its use as starter cultures in fermented foods" is an interesting review of the literature in line with the trends of a new approach to food fermentation.

Due to changes in the yeast nomenclature, please check all names in the text very carefully. For example, C. pulcherrima should be replaced with Metschnikowia pulcherrima. Teleomorphic names rather than anamorphic forms of yeast are now recommended.

line 15 - twice is "beer"

Keywords should be chosen more carefully, selected types of Candida and selective food use are mentioned, please generalize.

In Chapter 1, the authors should mention C. auris, which is currently a huge health risk, and on the other hand, C. utilis (now Cyberlindnera jadinii), which is used in food and feed production.

Table 1 - tables should be ordered, e.g. in terms of Types or the medium used.

Only 57 literature items are from the last 5 years (which is less than 50% of all), authors should also add articles from recent years to their review.

The authors used the old manuscript form.

The advantage of the manuscript is that in order to use Candida yeast in food, it is necessary to thoroughly understand their characteristics and influence on the human health.

Author Response

Reviewer #1

Comments and Suggestions for Authors

The manuscript "What is Candida doing in my food? A review and safety alert on its use as starter cultures in fermented foods" is an interesting review of the literature in line with the trends of a new approach to food fermentation. Due to changes in the yeast nomenclature, please check all names in the text very carefully. For example, C. pulcherrima should be replaced with Metschnikowia pulcherrima. Teleomorphic names rather than anamorphic forms of yeast are now recommended. We appreciate your comments and suggestions.  Please check 19, 49, 136 and Figure 1.  

line 15 - twice is "beer" OK, done. Please check line 19.

Keywords should be chosen more carefully, selected types of Candida and selective food use are mentioned, please generalize. OK, done. Please check key words.

In Chapter 1, the authors should mention C. auris, which is currently a huge health risk, and on the other hand, C. utilis (now Cyberlindnera jadinii), which is used in food and feed production. OK, done. Please check line 53.

Table 1 - tables should be ordered, e.g. in terms of Types or the medium used. OK, done. The Table was ordered in terms of isolation source (clinical, food and environmental samples). Please check Table 1.

Only 57 literature items are from the last 5 years (which is less than 50% of all), authors should also add articles from recent years to their review. OK, done. Please check some novel references numbers [18],[40],[44],[46].

The authors used the old manuscript form. The advantage of the manuscript is that in order to use Candida yeast in food, it is necessary to thoroughly understand their characteristics and influence on the human health.

Reviewer 2 Report

This review paper summarizes the Candida yeasts with a view to their use in fermented foods. Although there are many review articles on pathogenic Candida yeasts, information on Candida yeasts involved in food production has not been well summarized, even though they are involved in the production of many fermented foods. This review summarizes well the Candida yeasts involved in food production. I suggest only a minor revision of the manuscript.

Line 48, Misspelling of “S. cerevisiae

Line 95, Only “6.0” is not unified in significant figures.

Line 111, I think “food” should be plural.

Lines 113-114, Degree Symbol underlines should be deleted.

Line 137, Saccharomyces should be abbreviated as S..

Line 137, I don't think a sequence of “in” is desirable.

Line 145, In lines 73-74, the authors described about 40 of the around 200 Candida species as human pathogens. 10% does not match this description.

Line 147, “GIT” is already in line 51.

Line 269, Correct b to beta.

Table 1, Text alignment is not uniform.

Table 2, The meaning of the arrows in the table is not understood.

Table 2, The boundary of each subject is not easy to understand.

Author Response

Reviewer #2

This review paper summarizes the Candida yeasts with a view to their use in fermented foods. Although there are many review articles on pathogenic Candida yeasts, information on Candida yeasts involved in food production has not been well summarized, even though they are involved in the production of many fermented foods. This review summarizes well the Candida yeasts involved in food production. I suggest only a minor revision of the manuscript.Line 48, Misspelling of “S. cerevisiaeWe appreciate your comments and suggestions.  Please check line 49.

Line 95, Only “6.0” is not unified in significant figures. OK, done. Please check line 96.

Line 111, I think “food” should be plural. OK, done. Please check line 112.

Lines 113-114, Degree Symbol underlines should be deleted. OK, done. Please check line 114-116.

Line 137, Saccharomyces should be abbreviated as S.. OK, done. Please check line 138.

Line 137, I don't think a sequence of “in” is desirable. OK, done. Please check line 138.

Line 145, In lines 73-74, the authors described about 40 of the around 200 Candida species as human pathogens. 10% does not match this description. OK, done. Please check line 147.

Line 147, “GIT” is already in line 51. OK, done. Please check line 149.

Line 269, Correct b to beta. OK, done. Please check line 264.

Table 1, Text alignment is not uniform. OK, done. Please check Table 1.

Table 2, The meaning of the arrows in the table is not understood. OK, done. Please check Table 2 legend in the bottom of the table.

Table 2, The boundary of each subject is not easy to understand. OK, done. Upper border of each section was added in order to facilitate the understanding. Please check sections of Table 2.

Reviewer 3 Report

The present review article presents an important assessment regarding the use of Candida in the food industry. Although it presents several positive characteristics, it can also produce several health-related concerns.

Thus, although the manuscript is important in the field, it still has to be significantly improved. The manuscript should be revised by a native English speaker, better organised, topographically corrected, and some terms should be better described.

line 15 - Beer appears twice in this sentence: vegetable, meat, beer, wine and beer

line 55 - public health (Rajkowska & Kunicka-StyczyÅ„ska, 2018).  - this reference is not correctly added. Please see the author's guidelines.

line 105 - Please define the abbreviation "G+C content of the DNA": guanine-cytosine content

lines 113,114, 115, etc - Please use the degree sign, and add a space between the number and degree sign. Revise the whole manuscript.

line 121-122 - This section needs to be supported by a reference: https://doi.org/10.3390/microorganisms7080265

line 147 - The "gastrointestinal tract " has already been abbreviated at line 51. After the first abbreviation please use in the whole manuscript the abbreviated form.

line 202 - NCAC - please define the abbreviation

one 206 - please define the term dysbiosis: 10.3389/fmed.2022.813204;  10.1038/nri.2017.55

line 224 - Please define the term eubiosis: 10.1155/2020/9560684; 10.1186/s40168-020-00875-0

line 229 - 232 - this phrase should be based on a recent study: i.e. regarding the formation of organic acids and pH decrease: 10.3390/biology11040553

table 1 -Please format the table and use each name under the same form: Sabouraud dextrose or SAB; CHROMagar Candida culture media or  CHROMagar Candida, and so on.

- In the table, it is sufficient to use the abbreviated form, and in the table footer should be put the definitions of the abbreviations.

- Also, or use the whole name Candida .... or use it as C. ...

- line 272 - please use the abbreviated form! (CHROMagar Candida) - revise each term in the manuscript.

- line 342-344 - The present study should be cited: https://doi.org/10.1111/jam.13548

- line 410 - Please correct ("through neighbor-joining method"): through the neighbour-joining method

After the implementation of these corrections, the manuscript can be considered for publication.

Author Response

Reviewer #3

Comments and Suggestions for Authors.

The present review article presents an important assessment regarding the use of Candida in the food industry. Although it presents several positive characteristics, it can also produce several health-related concerns. Thus, although the manuscript is important in the field, it still has to be significantly improved. The manuscript should be revised by a native English speaker, better organised, topographically corrected, and some terms should be better described.

We appreciate your comments and suggestions to improve the article. T Please see the changes, marked in red, on the following lines: 16, 38, 45, 49, 53, 76, 77, 136, 145, 161, 173, 184, 204, 224, 227, 237, 241, 245, 247, 283, 297, 305, 318, 319, 334, 347, 353, 376, and 390.

line 15 - Beer appears twice in this sentence: vegetable, meat, beer, wine and beer OK, done. Please check line 16.

line 55 - public health (Rajkowska & Kunicka-StyczyÅ„ska, 2018).  - this reference is not correctly added. Please see the author's guidelines. OK, done. Please check line 56.

line 105 - Please define the abbreviation "G+C content of the DNA": guanine-cytosine content OK, done. Please check line 106.

lines 113,114, 115, etc - Please use the degree sign, and add a space between the number and degree sign. Revise the whole manuscript. OK, done. Please check line 114-116.

line 121-122 - This section needs to be supported by a reference: https://doi.org/10.3390/microorganisms7080265 OK, done. Please check line 124.

line 147 - The "gastrointestinal tract " has already been abbreviated at line 51. After the first abbreviation please use in the whole manuscript the abbreviated form. OK, done. Please check line 52.

line 202 - NCAC - please define the abbreviation It is already defined in line 179. Please check line 178.

one 206 - please define the term dysbiosis: 10.3389/fmed.2022.813204;  10.1038/nri.2017.55 OK, done. Please check line 207.

line 224 - Please define the term eubiosis: 10.1155/2020/9560684; 10.1186/s40168-020-00875-0 OK, done. Please check line 224.

line 229 - 232 - this phrase should be based on a recent study: i.e. regarding the formation of organic acids and pH decrease: 10.3390/biology11040553 OK, done. Please check line 231.

table 1 -Please format the table and use each name under the same form: Sabouraud dextrose or SAB; CHROMagar Candida culture media or  CHROMagar Candida, and so on. OK, done. Please check Table 1.

- In the table, it is sufficient to use the abbreviated form, and in the table footer should be put the definitions of the abbreviations. OK, done. Please check Table 1 bottom legend.

- Also, or use the whole name Candida .... or use it as C. ... OK, done. Please check Table 1.

- line 272 - please use the abbreviated form! (CHROMagar Candida) - revise each term in the manuscript. OK, done. Please check line 281.

- line 342-344 - The present study should be cited: https://doi.org/10.1111/jam.13548 OK, done. Please check line 349.

- line 410 - Please correct ("through neighbor-joining method"): through the neighbour-joining method OK, done. Please check line 397.

After the implementation of these corrections, the manuscript can be considered for publication.

Round 2

Reviewer 1 Report

The manuscript has been significantly corrected, but there are still naming errors. There are typos in the names (Table 1, last line on the first page), sometimes the authors used a new nomenclature for a given species, and sometimes not - it has to be standardized. The authors have not checked all names and still some are named according to the old nomenclature (eg Candida guilliermondii). Error in line 53 - Cyberlindnera jadinii is the new name of C. utilis.

Author Response

Thank you for your observations. All typos in the names have been corrected according to current literature:

  • C. parapsilosis (reclassified as Monilia parapsilosis)  https://www.cureus.com/articles/31480-clavicular-osteomyelitis-secondary-to-candida-parapsilosis-infection
  •  
  • C. famata (reclassified as Debaryomyces hansenii) https://journals.asm.org/doi/epub/10.1128/JCM.01811-20
  •  
  • C. zeylanoides (reclassified as Kurtzmaniella zeylanoides) ; C. zeylanoides (reclassified as Kurtzmaniella zeylanoides); C. norvegensis (reclassified asPichia norvegensis); C. utilis (reclassified as Cyberlindnera jadinii); (C. krusei (reclassified as Pichia kudriavzevii), ; 
  • C. rugosa (reclassified as Diutina rugosa) and C. mesorugosa (reclassified as Diutina mesorugosa)   https://www.sciencedirect.com/science/article/pii/S0963996920305925?casa_token=5q8mkYmVG8MAAAAA:vYf4xvPKZR4tyAV8F5P6yCNk_fZHk5oMv6a8wC_jpDZi73jBTANAswoTZFCs5w91J6b5THxmqg 
  • C. zemplinina (synonym to Starmerella bacillaris) https://link.springer.com/article/10.1007/s00253-020-11041-9
  •  
  • Candida amapae (reclassified as Saccharomycopsis amapae) https://www.microbiologyresearch.org/content/journal/ijsem/10.1099/ijsem.0.003000
  •  
  • Candida homilentoma (reclassified as Hyphopichia homilentoma), https://link.springer.com/article/10.1007/s00203-022-02811-2
  •  
  • C. zemplinina (reclassified as Starmerella bacillaris) https://www.sciencedirect.com/science/article/pii/S0924224422002084?casa_token=eZpr7MK-DCEAAAAA:NG-sNijbiWRm-nCERRBOFPna6MwhRNtNiXgc_Aq6FsgVgN1cJz_Mg2o5WaiZH6hOSNTjFe58EQ
  •  
  • C. membranifaciens (reclassified as Candida flareri) https://www.mdpi.com/2309-608X/8/3/254 
  •  
  • Candida magnoliae (reclassified as Starmerella magnoliae) https://www.mdpi.com/2076-2607/8/11/1789

Reviewer 3 Report

There authors considerably improved the manuscript, although in table 1 there are still some abbreviations that appear in both the abbreviated and whole form: CAC = CHROMagar Candida. Also at first use in the text this abbreviation should be defined. 

After these corrections the manuscript can be accepted for publication.

Author Response

Thank you for your comments. Table 1 and names throughout the manuscripts have been corrected according to your observations.